# Large magneto-Seebeck effect in magnetic tunnel junctions with half-metallic Heusler electrodes

Alexander Boehnke[1], Ulrike Martens[2], Christian Sterwerf[1], Alessia Niesen[1], Torsten Huebner[1], Marvin von der Ehe[2], Markus Meinert [1], Timo Kuschel[1,3], Andy Thomas[4], Christian Heiliger[5], Markus Münzenberg [2] & Günter Reiss [1]

Spin caloritronics studies the interplay between charge-, heat- and spin-currents, which are initiated by temperature gradients in magnetic nanostructures. A plethora of new phenomena has been discovered that promises, e.g., to make wasted heat in electronic devices useable or to provide new read-out mechanisms for information. However, only few materials have been studied so far with Seebeck voltages of only some microvolt, which hampers applications. Here, we demonstrate that half-metallic Heusler compounds are hot candidates for enhancing spin-dependent thermoelectric effects. This becomes evident when considering the asymmetry of the spin-split density of electronic states around the Fermi level that determines the spin-dependent thermoelectric transport in magnetic tunnel junctions. We identify $Co_2FeAl$ and $Co_2FeSi$ Heusler compounds as ideal due to their energy gaps in the minority density of states, and demonstrate devices with substantially larger Seebeck voltages and tunnel magneto-Seebeck effect ratios than the commonly used Co-Fe-B-based junctions.

[1] Center for Spinelectronic Materials and Devices, Physics Department, Bielefeld University, Universitätsstr. 25, D-33615 Bielefeld, Germany. [2] Institut für Physik, Ernst-Moritz-Arndt-Universität, Felix-Hausdorff-Str. 6, 17489 Greifswald, Germany. [3] Physics of Nanodevices, Zernike Institute for Advanced Materials, University of Groningen, Nijenborgh 4, 9747 AG Groningen, The Netherlands. [4] Leibniz Institute for Solid State and Materials Research Dresden (IFW Dresden), Institute for Metallic Materials, Helmholtzstrasse 20, 01069 Dresden, Germany. [5] Institut für theoretische Physik, Justus-Liebig-Universität Gießen, Heinrich-Buff-Ring 16, 35392 Gießen, Germany. Correspondence and requests for materials should be addressed to T.K. (email: tkuschel@physik.uni-bielefeld.de) or to G.R. (email: reiss@physik.uni-bielefeld.de)

The search for new materials and phenomena that enable, e.g., energy efficient sensors or memories, is a major driver for research in magnetism. Particularly, the emerging field of spin caloritronics[1, 2] combines spintronics and thermoelectrics, and provides a variety of new effects that might enable waste heat recovery in memory and sensor devices. One of the basic phenomena is the tunnel magneto-Seebeck effect (TMS)[3, 4], i.e., the change of the Seebeck coefficients of a magnetic tunnel junction (MTJ) when switching between parallel (P) and antiparallel (AP) magnetization alignment. An important benefit of the TMS effect is its occurrence in MTJs. These spintronic devices are available in high quality, facilitating the implementation of the TMS into future electronics. Despite these benefits, only low TMS ratios of a few 10% and Seebeck voltages in the microvolt range have been obtained so far for MTJs with Co–Fe–B or Co–Fe electrodes[3–11].

While these types of MTJs form the backbone of modern spintronics, due to their high tunnel magnetoresistance (TMR) effect of several hundred percent[12], they do not provide similarly large TMS ratios. This makes the increase of the TMS a challenging and important task for material research. Furthermore, the connection between the TMR and TMS effects is a fundamental question[13–15]. It is predicted theoretically that the sizes of the TMR and TMS effects are not directly linked and depend on different features of the density of states (DOS) of the ferromagnetic electrodes. Theory even predicts that large TMS effects can be achieved in MTJs that exhibit no TMR[14]. However, an experimental proof has been lacking so far.

In this paper, we show that the TMS is significantly enhanced by replacing conventional Co-Fe-based ferromagnets (FM) by nearly half-metallic (HM) Heusler compounds[16–22]. We first introduce an approach to describe the thermoelectric transport in an MTJ based on the DOS of the electrode material. This simple model is a powerful tool for identifying the important parameters, e.g., the shape of the DOS and the position of the chemical potential, that a material should possess to enable high Seebeck voltages and high TMS effect ratios. Similar results have been found by *ab initio* calculations[15, 23]. Based on these predictions, we experimentally investigate the two nearly HM Heusler compounds $Co_2FeSi$ and $Co_2FeAl$, and finally compare the experimental results obtained from the Heusler compound-based MTJs to Co-Fe based MTJs. The results prove a significantly enhanced TMS effect for MTJs with nearly HM Heusler compounds and that the size of the TMR and TMS are not directly correlated.

## Results

**Theoretical model.** For obtaining a basic understanding of the charge transport in an MTJ it is most convenient to examine the electronic DOS of the electrodes and their occupation (Fig. 1a–c). In this picture, a temperature difference $\Delta T$ results in different broadenings of the occupations in the hot and cold electrode according to the Fermi-Dirac distribution. This difference in occupation generates diffusion currents between occupied (dark color) and unoccupied states (bright color). Accordingly, above the chemical potential $\mu$ electrons travel from the hot to the cold electrode, while below $\mu$ electrons travel conversely from the cold to the hot side.

For an MTJ containing two conventional FMs in the free-electron picture (DOS(E) $\propto \sqrt{E}$, Fig. 1a) these opposing currents are of similar sizes, because a similar number of states is available above and below $\mu$ due to the flat DOS ($\partial DOS(E)/\partial E$ is small) in proximity to $\mu$. Only the tunneling probability is enhanced for electrons at higher energies, causing slightly more electrons to travel above $\mu$ than below. Still, the net diffusion across the barrier and, hence, the expected Seebeck voltage $V = -S\Delta T$ ($S$: Seebeck coefficient) is small for this type of MTJ. Since this is

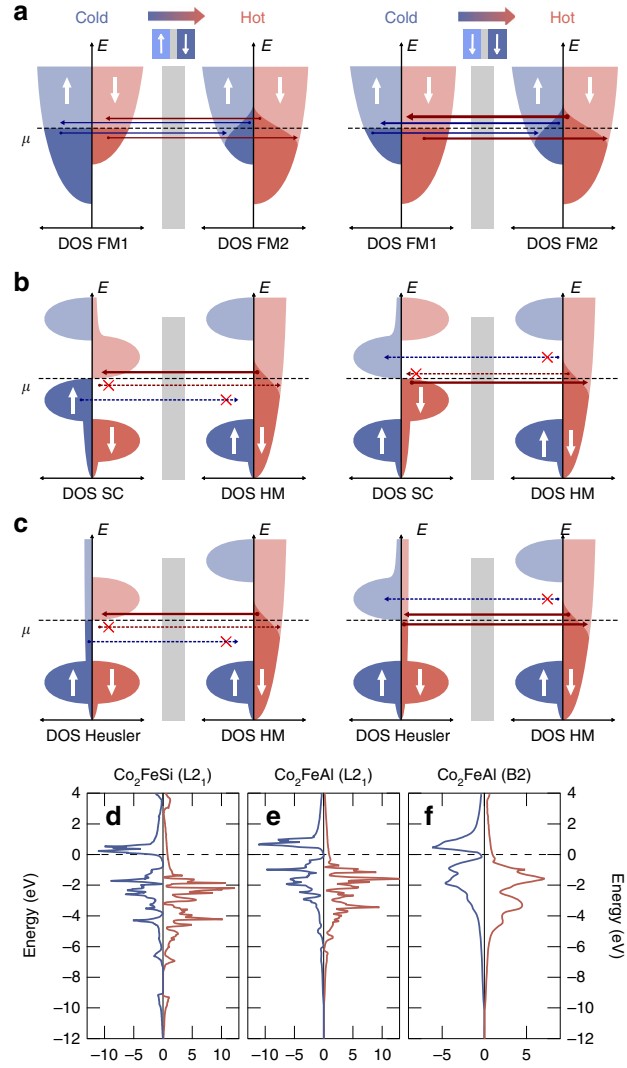

**Fig. 1** Influence of the DOS. AP and P state. **a**, FM I FM. Since the individual currents cancel out, the Seebeck effect and TMS ratio are small. **b**, FMSC I HM. In the AP state electrons only travel above $\mu$. In the P state electrons only travel below $\mu$. This is optimal for generating large Seebeck coefficients with opposite sign. **c**, Heusler I HM. In the AP state the transport is similar to the FMSC I HM MTJ. In the P state electrons travel above and below $\mu$. Hence, the Seebeck effect in the P state is small, while it is large in the AP state, leading to a large TMS ratio. **d–f**, The DOS obtained from DFT for Heusler compounds with theoretically predicted high TMS

valid for the P and the AP state of the MTJ, both Seebeck coefficients $S_P$ and $S_{AP}$ have similarly small values. Hence, the TMS ratio

$$TMS = \frac{S_P - S_{AP}}{\min(|S_P|, |S_{AP}|)} \quad (1)$$

is expected to be small in this type of MTJ.

For increasing the Seebeck effect, one of the diffusion channels above or below $\mu$ has to be suppressed. This is achieved by introducing a gap similar to semiconductors (SC). Conventional SCs have already proven to be useful in spin caloritronic applications[24–26]. In SCs the size and sign of $S$ are defined by the position of $\mu$ within the gap. $S$ is positive, if $\mu$ is located at the bottom edge of the gap, i.e, a p-type SC, whereas $S$ is negative, if $\mu$ is located at the top edge of the gap, i.e., an n-type SC. Hence, for

gaining a large difference between $S_P$ and $S_{AP}$ it is desired to magnetically switch between these two types of SCs. The DOS of a ferromagnetic semiconductor (FMSC) depicted for the left cold electrode in Fig. 1b allows this switching. In the FMSC spin-up electrons (blue) occupy a p-type DOS, while spin-down electrons occupy an n-type DOS. For selecting, which of these electrons contribute to the transport, we choose a HM with a gap in the spin-up states and a metallic DOS for the spin-down electrons as a counter electrode. Since the gap is larger than the thermal activation energy in the hot HM, only the metallic spin-down DOS contributes to the transport, resulting in an n-type like behavior of the junction and, hence, a large positive $S$. However, when the magnetization of the FMSC is reversed, the spin-down states that contribute to the tunneling process exhibit a p-type like DOS and, hence, we receive a large negative $S$. Clearly, this would be perfect for reading the two states of the MTJ by the Seebeck voltage.

Since experimentally realizing FMSCs is very challenging, it is desirable to find another more accessible material class that also fulfills the properties of an FMSC | HM MTJ. Heusler compounds are a very promising substitute for the FMSC (Fig. 1c), since some of them, such as $Co_2FeSi$ (Fig. 1d), also reveal an n-type gap in the spin-down DOS. However, they do not possess a p-type gap, but a metallic DOS in the spin-up channel. Thus, when combined with a HM, we only achieve a switching between an n-type like transport in the AP state and a metallic behavior in the P state of the MTJ (Fig. 1c). Our model therefore predicts a large positive Seebeck coefficient, i.e., a negative Seebeck voltage ($V = -S\Delta T$), in the AP state and a strongly reduced Seebeck effect in the P state. Hence, we expect that a Heusler | HM MTJ simultaneously provides a high TMS ratio and a large Seebeck voltage in the AP state.

For experimentally realizing this device, we use the Heusler compounds $Co_2FeSi$ and $Co_2FeAl$ in combination with an MgO barrier and a Co-Fe-based counter electrode. Co-Fe on its own is not a HM, but when combined with a crystalline MgO barrier the tunneling process becomes coherent. Under these conditions, the electrons contributing to the tunneling process reveal a HM nature[27]. For checking the properties of the Heusler compounds $Co_2FeSi$ and $Co_2FeAl$, we have performed density functional theory (DFT) calculations[21] of the spin-resolved DOS (Fig. 1d–f). In the minority DOS, $Co_2FeSi$ has a pseudogap right below the chemical potential, while a large number of unoccupied states is found directly above $\mu$. On the contrary, the majority DOS is rather flat around $\mu$. Hence, this DOS perfectly resembles the ideal DOS for a high TMS as sketched in Fig. 1c. $Co_2FeAl$ in the full-Heusler $L2_1$ ordering is predicted to have a pseudogap that is positioned relatively symmetrical around $\mu$ (Fig. 1f). Accordingly, a much smaller Seebeck effect is expected. However, $Co_2FeAl$ does not crystalize in the full-Heusler structure, but energetically prefers the less-ordered $B2$ ordering, which results in a less pronounced gap. Additionally, the large number of states above $\mu$ is shifted closer towards $\mu$. Thus, similar TMS effects are expected for $Co_2FeSi$ in the $L2_1$ ordering and $Co_2FeAl$ in the $B2$ ordering. Only the absolute Seebeck coefficients are expected to be slightly larger for $Co_2FeSi$ due to the wider gap and larger asymmetry.

To experimentally explore the tunneling properties, we performed tunneling spectroscopy with d$I$/d$V$ measurements (see Supplementary Note 5 including ref. [28]). The data show that none of the used MTJs show a pure parabolic dependence of d$I$/d$V$ on the voltage $V$, revealing either coherent tunneling for the CoFeB case or strong deviations of the DOS from an s-like band with DOS$(E) \propto \sqrt{E}$. Furthermore, X-ray diffraction, atomic force microscopy, and X-ray fluorescence have been used to proof the excellent quality of our samples (see Supplementary Note 2 including ref. [29]). Since the structural results, the TMR values,

and the d$I$/d$V$ curves are quite comparable to earlier work by Mann et al.[30], we conclude that our samples have similar half metallic character as shown by inelastic tunneling spectroscopy and ultrafast demagnetization experiments on their samples.

**Experimentally determined TMS effects**. For experimentally determining the thermoelectric properties of the MTJs, we generate a temperature gradient across the barrier and record the Seebeck voltages for the two magnetic configurations of the MTJ. For heating we use a diode laser (wavelength 635 nm) that is modulated on/off at a frequency of 13 Hz for the high-impedance Heusler MTJs and at 1.5 kHz for the lower-impedance Co-Fe-B MTJs. The beam is focussed on top of a gold transducer placed above the MTJ. The spot size on the transducer is adjusted according to the size of the MTJs to guarantee a homogenous heating. The generated Seebeck voltages are amplified by a high-impedance amplifier and afterward fed into a lock-in amplifier set to voltage mode. Thus, all Seebeck voltages are given as the effective values of the first harmonic of the lock-in amplifier. In a DC experiment the values are expected to be twice as high[5]. For time-dependent investigations, e.g., checking the saturation of the signal, the voltages are recorded with an oscilloscope (see also Supplementary Note 3). We have published a detailed description of the setup used in this work in ref. [5].

**TMS in $Co_2FeSi$ MTJs**. The first Heusler compound discussed here is $Co_2FeSi$[31]. The complete MTJ stack consists of MgO (substrate)/MgO (5)/Cr (5)/$Co_2FeSi$ (20)/MgO (2)/$Co_{70}Fe_{30}$ (5)/$Mn_{83}Ir_{17}$ (10)/Ru (25), numbers indicate thicknesses in nm (cf. Supplementary Note 1). By e-beam patterning, MTJ pillars are produced, such that the Cr and half of the $Co_2FeSi$ layer remain as bottom lead. After insulating the MTJs with $Ta_2O_5$, contact pads consisting of Ta (5)/Au (60) are placed on top of the MTJs to allow electrical contact. Furthermore, these pads assure that the laser is fully absorbed by the Au layer and only the heat is transferred to the functional layers of the MTJ.

Figure 2a displays the Seebeck voltage of a $Co_2FeSi$ MTJ for different heating powers in dependence on an external magnetic field. The characteristic minor-loop of an exchanged-biased MTJ is clearly recognizable. Moreover, the MTJ exhibits a nearly identical switching behavior of the Seebeck voltage and the resistance (Fig. 2b, see also Supplementary Note 3). We further detected the same behavior when directly measuring the Seebeck current (cf. Supplementary Note 4). This is expected, because in the TMS as well as in the TMR effect the spin-dependent transport is altered by changing the magnetization alignment of the FM layers of the MTJ. The Seebeck voltage rises linearly with applied laser power, i.e., enhanced temperature difference across the barrier (Fig. 2c). The largest values reached in our experiments are $V_{AP} \approx -664\,\mu V$ and $V_P \approx -370\,\mu V$ at 150 mW heating power. This yields a TMS ratio of $-80\%$. For lower heating powers the absolute TMS effect ratio is even enhanced, reaching its highest value of $-86\%$ at 10 mW heating power. The base temperatures achieved by the laser heating reach from 300 K at 10 mW to 351 K at 150 mW (cf. Supplementary Note 7 including refs [32–41]). This trend shows that large TMS ratios and Seebeck voltages are observed at various heating powers, i.e., different temperature gradients and base temperatures.

To ensure that the spin-dependent thermovoltages are only generated by the MTJ and not by the FM bottom lead, we have forced the MTJ into a dielectric breakdown by applying a bias voltage of 4 V. The MTJ does not show any TMR after this treatment. The subsequently determined Seebeck voltages at unchanged irradiation conditions only reach $-2.2\,\mu V$ at 150 mW heating power. Furthermore, the dependence of the Seebeck

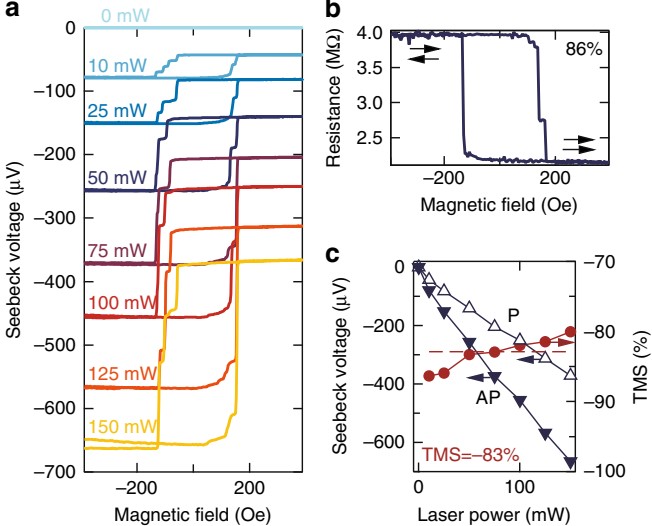

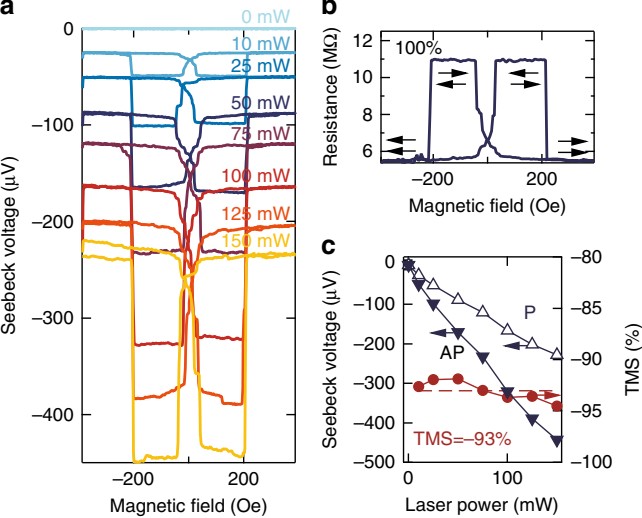

**Fig. 2** TMS effect in $Co_2FeSi$ based MTJs. **a**, Seebeck voltage in dependence on magnetic field for different laser powers. **b**, Minor loop of the TMR effect (resistance vs. magnetic field) of the same MTJ without laser heating. **c**, Power dependence of the Seebeck voltage and the TMS ratio extracted from the saturation voltages in **a**. The average TMS effect of the $Co_2FeSi$ based MTJs amounts to −83%

**Fig. 3** TMS effect in $Co_2FeAl$ based MTJs. **a** Seebeck voltage in dependence on magnetic field for different laser powers. **b** Major loop of the TMR effect (resistance vs. magnetic field) of the same MTJ without laser heating. **c** Power dependence of the Seebeck voltage and the TMS ratio extracted from the saturation voltages in **a**. The average TMS effect of the $Co_2FeAl$ based MTJs amounts to −93%

voltage on the external magnetic field vanishes (cf. Supplementary Note 6). Hence, we can fully attribute the high TMS to the intact Heusler based MTJ.

**TMS in $Co_2FeAl$ MTJs**. The layer stacks for the $Co_2FeAl$ based MTJs consist of MgO (substrate)/TiN (20)/$Co_2FeAl$ (10)/MgO (2)/$Co_{40}Fe_{40}B_{20}$ (5)/Ta(3)/Ru (3)[42] (cf. Supplementary Note 1). In these samples only the TiN layer serves as a common bottom lead. Accordingly, FM materials only remain in the MTJ pillars. The choice of an insulating substrate and a non- FM lead ensures that a response of the Seebeck effect to an external magnetic field has its origin solely in the MTJ pillars.

Figure 3a displays the Seebeck voltage of a $Co_2FeAl$ MTJ for different heating powers in dependence on an external magnetic field. The characteristic bow-tie shaped switching of a pseudo-spin-valve is clearly recognizable. Again, the MTJ shows a nearly identical switching of the Seebeck voltage and the resistance (Fig. 3b, see also Supplementary Note 3), and the Seebeck voltage rises linearly with applied laser power. Once more, a similar behavior has been observed when directly measuring the Seebeck current (cf. Supplementary Note 4). A TMS of −93 ± 2% is found for all applied heating powers, i.e, these $Co_2FeAl$ based MTJs show a nearly constant TMS over a wide range of base temperatures. For the largest heating power of 150 mW, the Seebeck voltages reach $V_{AP} = −442\,\mu V$ and $V_P = −227\,\mu V$ yielding a switching ratio of nearly −95%.

The small variation of the TMS ratio with laser power in these $Co_2FeSi$ and $Co_2FeAl$ based MTJs indicates a correspondingly small influence of the base temperature. This apparently weak temperature dependence is a favorable property for possible applications of the TMS such as a read-out of the information stored in the magnetic state of the MTJ.

Although we do not expect any influence of the TiN bottom lead on the switching of the Seebeck voltage with magnetic field, we also investigated the Seebeck voltage in the $Co_2FeAl$ based MTJs after dielectric breakdown. Similarly to the $Co_2FeSi$ based MTJs no switching of the Seebeck voltage is observed for the broken MTJs. Moreover, the absolute value of the Seebeck voltage is strongly decreased and even reverses its sign. For a heating

power of 150 mW we observe a Seebeck voltage of 15 μV in the broken MTJ. This remaining signal is probably generated by an in plane temperature gradient in the TiN.

**Comparison to Co–Fe–B MTJs**. A direct comparison of $Co_2FeAl$ and $Co_{26}Fe_{54}B_{20}$ based MTJs makes the benefits of using Heusler compounds for TMS devices obvious (Fig. 4a). The absolute value of the TMS ratio in $Co_2FeAl$ is nearly twice as high as in the Co-Fe-B-based MTJs. However, when comparing the TMR of the two materials we find exactly the opposite (Fig. 4b). The different effect sizes clearly reveal the different influence of the DOS on the two effects, and point out that MTJs with a high TMR do not necessarily generate a high TMS. This has already been predicted theoretically[13–15], but a reliable experimental prove had been missing up to now.

The favorable properties of the Heusler compound MTJs become even more clear when comparing multiple MTJs with $Co_2FeAl$ and $Co_2FeSi$ electrodes to MTJs containing different Co-Fe-B (Fig. 4c–e) compositions. The Heusler based MTJs do not only show higher TMS ratios of −80 to −120%, they also generate a substantially larger Seebeck voltage of up to −664 μV. For the Co-Fe-B-based MTJs this combination of a high Seebeck signal and a high TMS ratio cannot be observed. Although, the MTJs with two $Co_{26}Fe_{54}B_{20}$ electrodes generate TMS ratios of nearly 50% they cannot provide Seebeck voltages of more than 10 μV. For the $Co_{40}Fe_{40}B_{20}$ the Seebeck voltages are slightly increased and reach up to nearly 50 μV, but these samples only yield TMS ratios of maximum 10%. These observations for different Co-Fe compositions are consistent with *ab initio* calculations by Heiliger et al.[14]. Furthermore, Fig. 4e supports the prediction that the TMS and TMR ratios in MTJs with different DOS, i.e., electrode materials[13–15], are not directly correlated. High TMS effects are not observed in the MTJs with the highest TMR. A comparison of the Heusler- to the $Co_{40}Fe_{40}B_{20}$ based MTJs reveals that samples with similar TMR can possess drastically different TMS ratios. However, all MTJs containing Heusler compounds reveal simultaneously large Seebeck voltages and TMS effects, indicating the robustness of the effect size in these MTJs.

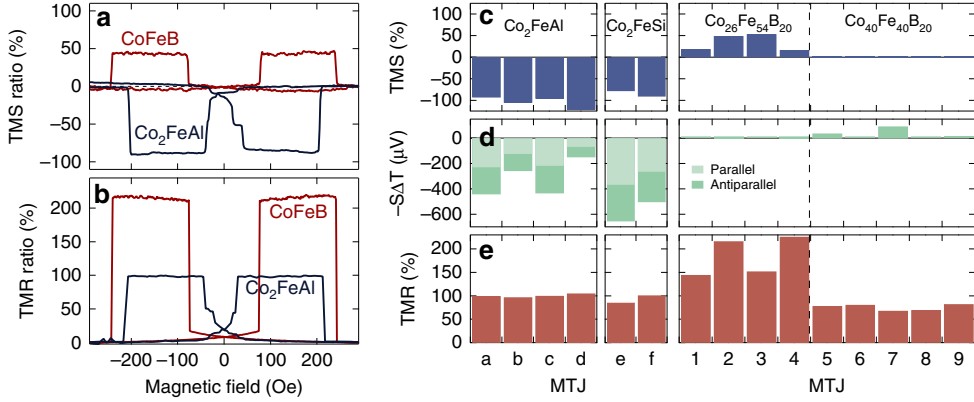

**Fig. 4** Comparison of Heusler compound to Co-Fe-B MTJs. **a** The TMS ratios (determined at 150 mW laser power) of $Co_2FeAl$ is twice as high compared to $Co_{26}Fe_{54}B_{20}$ and possesses an inversed sign. **b** For the TMR ratios (at 10 mV bias voltage) of the same MTJs the opposite is found. **c** TMS ratios for various $Co_2FeAl$ and $Co_2FeSi$ based MTJs in comparison to MTJs with two different Co-Fe-B compositions. **d** Corresponding Seebeck-voltages $-S\Delta T$ at a laser power of 150 mW in the P and AP configuration of the MTJs. **e** TMR ratio of the MTJs at 10 mV bias. Only the Heusler based MTJs simultaneously exhibit a large Seebeck voltage and high TMS ratio

---

**Table 1 Seebeck coefficients of MTJs with different materials**

| Materials | $V_P(\mu V)$ | $V_{AP}(\mu V)$ | $S_P(\mu VK^{-1})$ | $S_{AP}(\mu VK^{-1})$ | TMS |
|---|---|---|---|---|---|
| $Co_{40}Fe_{40}B_{20}$[a] | 90.6 | 93.2 | −750 | −770 | 2.8% |
| $Co_{26}Fe_{54}B_{20}$[b] | 6.0 | 9.0 | −545 | −818 | 50% |
| $Co_2FeAl$[c] | −227 | −442 | 582 | 1133 | −95% |
| $Co_2FeSi$[d] | −370 | −664 | 948 | 1703 | −80% |

The elliptical MTJs have diameters of 2 μm × 1 μm. The Seebeck voltages are obtained at 150 mW laser heating
[a]$Co_{40}Fe_{40}B_{20}$/MgO 1.5 nm/$Co_{40}Fe_{40}B_{20}$, for more values see ref. [5]
[b]$Co_{26}Fe_{54}B_{20}$/MgO 1.7 nm/$Co_{26}Fe_{54}B_{20}$, for more values see refs [3, 6]
[c]$Co_2FeAl$/MgO 2 nm/$Co_{40}Fe_{40}B_{20}$
[d]$Co_2FeSi$/MgO 2 nm/$Co_{70}Fe_{30}$

---

Finally, we estimate the Seebeck coefficients of the MTJs for enabling a quantitative comparison of their properties. As known from literature[3–6], the necessary determination of the temperature difference between the upper and the lower electrode of the MTJ is the most challenging aspect. Here, the temperature profile of each sample type is simulated by finite element methods (cf. Supplementary Note 7), and the Seebeck coefficient $S = -V/\Delta T$ is calculated from the Seebeck voltage and the temperature difference between the electrodes (Table 1). Again, a larger value of $S$ is found for the Heusler compound based MTJs. Furthermore, the sign of the Seebeck voltage and coefficient are reversed, when replacing Co-Fe-B by a Heusler compound. We find $S_{AP}$ up to 1703 μVK$^{-1}$ for the $Co_2FeSi$ based MTJ, while for the Co-Fe-B-based MTJ we only find −818 μVK$^{-1}$. Nonetheless, we would like to emphasize that the temperature differences determined by finite element methods can be quite inaccurate. Particularly, the thermal conductivity of the nanometer-thick MgO barrier is a subject discussed in literature[3, 9, 37, 43]. Hence, the presented Seebeck coefficients (Table 1) can only be compared to values obtained with the same method, e.g., in refs [3, 5, 6,], while the absolute values should be considered with great care.

For applications, however, the exact value of the Seebeck coefficient is of secondary interest. A large Seebeck voltage, as well as a high and stable readout contrast between the P and the AP state that can be used for further signal processing are of larger importance. With our direct comparison of Heusler- and Co-Fe-B-based MTJs, we have shown that MTJs based on Heusler compounds are able to provide a favorable combination of TMS ratios between −80 and −120% and large Seebeck voltages of up to −664 μV.

**Conclusion.** In conclusion, we have presented an experimental study of the spin-dependent Seebeck effect in Heusler compound based MTJs. Replacing one of the commonly used Co-Fe-B electrodes by a HM Heusler compound largely increases the TMS ratio from several percent up to −120% and simultaneously provides Seebeck voltages of up to 664 μV. These experimental findings can be explained based on the DOS. We have introduced a simple model that allows estimating the thermal diffusion currents, and, hence, the Seebeck coefficients, only from the DOS. This model does not only explain the observed effect sizes, but it is a powerful tool for quickly screening the capability of other materials for providing high TMS based on their DOS, available from material repositories, such as AFLOW.lib[44]. Thus, the more sophisticated and time-consuming *ab initio* studies can be concentrated on the most promising candidate materials. Since the model has already proven its usefulness in our experiments with Heusler compounds, we expect it to be helpful in finding suitable materials for thermoelectric effects and spin caloritronic transport phenomena.

Furthermore, our discovery of a material that enables high TMS ratios paves the way to a large number of new effects. The combination of the high TMS effect, high spin-polarization, and low Gilbert damping[45] makes these Heusler compound MTJs ideal for investigating the thermal spin transfer torque in MTJs[46–48], which is a key ingredient in the design of spin caloritronic memory devices. The high Seebeck effect might also enhance the magneto-Peltier effect that has been investigated in Co-Fe-B-based MTJs[49]. Moreover, our results might also be of interest for the future research on the heat distribution and the resulting effects in magnetic access memory devices (MRAM)[50, 51].

## Methods

For a detailed explanation of the MTJ preparation, characterization of the MTJs after dielectric breakdown and details on the finite element method simulations please refer to the Supplementary Information and refs [5,31,42].

**Data availability**. The data that support the findings of this study are available from the corresponding author upon request.

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

## Acknowledgements

We gratefully acknowledge financial support by the Deutsche Forschungsgemeinschaft (DFG) within the priority program SpinCaT SPP1538 (RE 1052/24–2, MU 1780/8–2, HE 5922/4–2, KU 3271/1–1, TH 1399/4–2) and the Bundesministerium für Bildung und Forschung (BMBF). Furthermore, we thank Benjamin Geisler for sharing his knowledge about Seebeck effects in half-metallic Heusler compounds.

## Author contributions

A.B., U.M., T.H. and M.v.d.E. set up the experiments and performed the measurements under supervision of T.K., G.R. and M.Mü. A.B., C.S. and A.N. prepared and characterized the samples. A.B., C.H. and G.R. developed und discussed the theoretical

model. M.Me. provided the DFT calculations. A.B. wrote the manuscript. A.T., C.H., M.Mü. and G.R. designed the research approach. All authors discussed the measurements and the manuscript.

## Additional information

**Competing interests:** The authors declare no competing financial interests.

