## [Peer Review File · Nature Communications]

Reviewers' comments:

Reviewer #1 (Remarks to the Author):

After thoroughly reading the paper, I am not totally convinced just yet that the paper warrants publication in Nature Communications in its present form.

This is a nice paper, where the authors argue that magnetic tunnel junctions (MTJs) formed from conventional Co-Fe or Co-Fe-B electrodes exhibit small Seebeck Voltages and small Tunneling Magneto-Seebeck (TMS) ratios, even though they exhibit large Tunneling Magneto Resistance (TMR). They suggest an alternative based on half-metallic Heusler electrodes, and experimentally demonstrate larger Seebeck voltages and their respective TMS ratios for Co₂FeAl (CFA) and Co₂FeSi (CFS) Heusler compounds, and present a comparison with conventional CoFeB based devices. To get this paper accepted, please consider adding these details:

1. Please show the dI/dV measurements across the Fermi Energy in both parallel and anti-parallel configurations for all the MTJs. If space is a problem, please add this in the supplementary. A comparison for the Heusler and conventional electrodes is a must.
2. X-TEM and composition analysis of Heusler alloys in thin film forms is desired.
3. Why is the TMR of a Heusler | MgO | CoFe not higher at RT? Based on the 2nm thick MgO, I would expect the TMR to be >200% (several papers from Japan have already reported this). Better material optimization is needed? Therefore, I am interested in the dI/dV measurements (point 1 above).
4. Please reference work from Jansen's group (Nature 2012, PRB 2015) and from Parkin's group (PNAS 2015). Spin accumulation voltage in Si, a semiconductor, is in the same range 100s of uV. Seebeck voltages for Parkin's paper is ~60uV for conventional materials, TMS ~10%, consistent with the author's claims. What the authors discuss is an extension to these prior reports as well as several of their own.
5. In Fig 1b, the authors have considered the possibility of a ferromagnetic semiconductor (FMSC) |MgO | half metal (HM) tunnel junction, where the sign of the Seebeck voltage in parallel and anti-parallel configurations of an MTJ will be opposite. This would allow for a substantially large TMS. But, they have not demonstrated this effect, and I don't expect them to either. However, this effect need not only be present in only such a scenario. The opposite sign of the Seebeck voltage for the parallel and anti-parallel configurations can be obtained even in conventional magnetic electrodes but, with selective spin filtering of the tunnel barrier, which is

difficult to control. I agree with the authors' claim that simultaneous presence of a large Seebeck voltage and a large TMS can only be realized in Heusler based electrodes. But, a TMS > TMR scenario is perfectly realizable, even though the actual Seebeck voltages are small.

Having said this, the overall work is very detailed and credit worthy. Congratulations on the hard work! If the authors could answer the above 5 points (please note that these are not objections), the paper would be stronger and would be worthy of publication in Nature Communications.

Reviewer #2 (Remarks to the Author):

The manuscript reports on spin-caloric measurements of magnetic tunnel junctions comprising a Heussler alloy, with a conventional ferromagnetic metal. Two Heussler alloys were investigated. The authors give qualitative reasons why one expects a larger spin Seebeck signal, as well as larger thermal spin valve signal--tunneling magnetic Seebeck effect, in such tunnel junctions, based on DFT supported density of states analysis. This is all reasonable and indeed well supported by experiment. The signals are indeed huge, of order 100 micro Vs, and the magnetic Seebeck effect is also large. The paper is very well written, the phenomena are rather systematically investigated, and the results are of great interest to the spin calorics community. I fully support its publication, having no major issues or comments.

Reviewer #3 (Remarks to the Author):

This paper reports a large spin-dependent thermoelectric effects from Heusler materials based magnetic tunnel junctions (MTJs) thus demonstrating that their Seebeck voltages and tunnel-Seebeck effect ratios are significantly larger than Co-Fe-B based MTJs. The authors attributed their findings to the unique band structure of Heusler materials showing good agreement with their density of states together with half-metallic counterpart electrode. Spin-caloritronics is still an active topic in spintronic and magnetic communities since possible high efficiency of spin current by thermoelectric effect may even switch the magnetization at low power consumption compared to spin-transfer torque switching although some underlying physics is still at large. In this sense, this work is of some interest possibly providing an important bridge to switching by pure thermal spin current only in the absence of external fields. The paper is well rewritten and clear. So I do not find any major issues that prevent publication of this paper in Nature Communications except some minor things as followings.

- (1) Some typos are located, e.g., Co₂₆F₅₄B₂₀ in second line of Table I, which I think should be Co₂₆Fe₅₄B₂₀.
- (2) Too many papers previously published from the same group seem to be self-referenced, and I

would suggest that this should be balanced. Also it is found that some papers were missing, e.g., Pushp et al, PNAS 112, 6585 (2015).

Response to Reviewers' Comments:

Responses to Reviewer #1 (Remarks to the Author):

Thanks to reviewer #1 for the thorough corrections of our paper and the helpful remarks. We appreciate his honesty and agree that considering the issues raised in the report of reviewer #1 will result in an improved manuscript. In the following, we will cite reviewer #1's comments (italic) and provide our answers or comments on the changes we have made to the manuscript.

After thoroughly reading the paper, I am not totally convinced just yet that the paper warrants publication in Nature Communications in its present form.

This is a nice paper, where the authors argue that magnetic tunnel junctions (MTJs) formed from conventional Co-Fe or Co-Fe-B electrodes exhibit small Seebeck Voltages and small Tunneling Magneto-Seebeck (TMS) ratios, even though they exhibit large Tunneling Magneto Resistance (TMR). They suggest an alternative based on half-metallic Heusler electrodes, and experimentally demonstrate larger Seebeck voltages and their respective TMS ratios for Co₂FeAl (CFA) and Co₂FeSi (CFS) Heusler compounds, and present a comparison with conventional CoFeB based devices. To get this paper accepted, please consider adding these details:

1. Please show the dI/dV measurements across the Fermi Energy in both parallel and anti-parallel configurations for all the MTJs. If space is a problem, please add this in the supplementary. A comparison for the Heusler and conventional electrodes is a must.

Thank you very much for pointing this out. The dI/dV measurements were added to the supplementary materials. We show dI/dV measurements in both parallel and anti-parallel configurations in dependence of the bias voltage for each MTJ type. All data clearly deviate from the parabolic shape that is expected for tunnelling between flat DOSs as described by Brinkman. Hence, we conclude that all MTJs possess a DOS with a gap. For a more detailed explanation, please refer to the corresponding section in the supplementary.

2. X-TEM and composition analysis of Heusler alloys in thin film forms is desired.

We do not have access to an X-TEM and such analysis of structure and composition would significantly delay the publication of the manuscript. However, we have included XRD characterization of the structure in the supplementary information. For determining the composition of the Co₂FeSi Heusler compounds XRF analysis with a resolution of about 1 at-% was used as discussed in one of our recent publications [C. Sterwerf et al. IEEE Trans. Magn. 49, 4386–4389 (2013)]. We fully agree that TEM analysis could add interesting information on the Heusler compounds' composition and structure. We are well aware that these properties significantly influence the electronic structure, e.g. the DOS of the Heusler compounds. In accordance with literature and to the best of our knowledge we have tried to take this into account by performing the DFT calculations based on the L2₁ structure for the Co₂FeSi and in the less ordered B2 phase for Co₂FeAl.

3. Why is the TMR of a Heusler | MgO | CoFe not higher at RT? Based on the 2nm thick MgO, I would expect the TMR to be >200% (several papers from Japan have already reported this). Better material optimization is needed? Therefore, I am interested in the dI/dV measurements (point 1 above).

The TMR is indeed smaller than in the work done in Japan. As far as we know, they have observed the highest TMR in significantly off-stoichiometric $\text{Co}_2\text{Mn}_{1+x}\text{Si}$ [H. X. Liu et al. Appl. Phys. Lett. 101, 132418 (2012)]. We, however, have used stoichiometric Heusler compounds with a relatively thick tunnel barrier. These differences between our samples and the Japanese samples can most probably explain the lower TMR for our samples by the presence of dislocations in the Heusler compound.

For the Co_2FeSi we are not aware of any higher TMR values, than those measured in the Bielefeld group by [C. Sterwerf et al. IEEE Trans. Magn. 49, 4386–4389 (2013)]. Since we have used layer stacks produced in the same sputtering tool at the same time as Sterwerf et al., only the different patterning that is needed for the TMS experiments under laser irradiation can be the origin for the slightly reduced TMR ratio.

4. Please reference work from Jansen's group (Nature 2012, PRB 2015) and from Parkin's group (PNAS 2015). Spin accumulation voltage in Si, a semiconductor, is in the same range 100s of μV . Seebeck voltages for Parkin's paper is $\sim 60\mu\text{V}$ for conventional materials, TMS $\sim 10\%$, consistent with the author's claims. What the authors discuss is an extension to these prior reports as well as several of their own.

References from Jansen's group have been added (1. K.-R. Jeon, H. Saito, S. Yuasa & R. Jansen Phys. Rev. B 92, 54403 (2015). 2. J. C. Le Breton, S. Sharma, H. Saito, S. Yuasa & R. Jansen Nature 475, 82–5 (2011). 3. R. Jansen Nat. Mater. 11, 400–8 (2012).)

We apologize for not including these publications in our first version. They are obviously important contribution to the work of spin caloritronics with semiconductors and fit extraordinarily well into our manuscript.

Furthermore, we have added the paper by Parkin's group that nicely underpins our experiments at Co-Fe-B based MTJs with MgO barrier. Additionally, we have included two recent publications from T. Böhnert et al. These also investigate CoFeB based MTJs. (1. T. Böhnert et al. Phys. Rev. B 95, 104441 (2017). 2. T. Böhnert et al. J. Phys. Condens. Matter 29, 185303 (2017).)

5. In Fig 1b, the authors have considered the possibility of a ferromagnetic semiconductor (FMSC) |MgO| half metal (HM) tunnel junction, where the sign of the Seebeck voltage in parallel and anti-parallel configurations of an MTJ will be opposite. This would allow for a substantially large TMS. But, they have not demonstrated this effect, and I don't expect them to either. However, this effect need not only be present in only such a scenario. The opposite sign of the Seebeck voltage for the parallel and anti-parallel configurations can be obtained even in conventional magnetic electrodes but, with selective spin filtering of the tunnel barrier, which is difficult to control. I agree with the authors' claim that simultaneous presence of a large Seebeck voltage and a large TMS can only be realized in Heusler based electrodes. But, a TMS > TMR scenario is perfectly realizable, even though the actual Seebeck voltages are small.

We fully agree with reviewer #1. Theory shows that it should even be possible to realize an MTJ that does not exhibit TMR, but that shows a high TMS. C. Heiliger already suggested this in his paper on the TMS effect [C. Heiliger et al. Phys. Rev. B 87, 224412 (2013)]. We have added a sentence about this possibility to the manuscript on page 1 line 55.

An experimental proof might be difficult, since not all needed material compositions are stable and, up to now, we have used the TMR to prove the switching of the Seebeck signal between p and ap

in the TMS measurement.

Having said this, the overall work is very detailed and credit worthy. Congratulations on the hard work! If the authors could answer the above 5 points (please note that these are not objections), the paper would be stronger and would be worthy of publication in Nature Communications.

We thank reviewer #1 for these friendly and motivating words as well as for the constructive feedback. We are very happy that our work is perceived in such a positive way.

Responses to Reviewer #2 (Remarks to the Author):

The manuscript reports on spin-caloric measurements of magnetic tunnel junctions comprising a Heussler alloy, with a conventional ferromagnetic metal. Two Heussler alloys were investigated. The authors give qualitative reasons why one expects a larger spin Seebeck signal, as well as larger thermal spin valve signal--tunneling magnetic Seebeck effect, in such tunnel junctions, based on DFT supported density of states analysis. This is all reasonable and indeed well supported by experiment. The signals are indeed huge, of order 100 micro Vs, and the magnetic Seebeck effect is also large. The paper is very well written, the phenomena are rather systematically investigated, and the results are of great interest to the spin calorics community. I fully support its publication, having no major issues or comments.

— We greatly appreciate this very positive feedback of reviewer #2 to our submitted manuscript. We are very happy that our work is perceived in such a positive way and that you are recommending it for publication in Nature Communications.

Responses to Reviewer #3 (Remarks to the Author):

We thank reviewer #3 for his positive feedback and we are happy that our findings are of great interest and might make an important contribution to the field of spin caloritronics. In the following reviewer #3's questions and comments (italic) are addressed (regular) in a point-by-point structure.

This paper reports a large spin-dependent thermoelectric effects from Heusler materials based magnetic tunnel junctions (MTJs) thus demonstrating that their Seebeck voltages and tunnel-Seebeck effect ratios are significantly larger than Co-Fe-B based MTJs. The authors attributed their findings to the unique band structure of Heusler materials showing good agreement with their density of states together with half-metallic counterpart electrode. Spin-caloritronics is still an active topic in spintronic and magnetic communities since possible high efficiency of spin current by thermoelectric effect may even switch the magnetization at low power consumption compared to spin-transfer torque switching although some underlying physics is still at large.

In this sense, this work is of some interest possibly providing an important bridge to switching by pure thermal spin current only in the absence of external fields. The paper is well rewritten and clear. So I do not find any major issues that prevent publication of this paper in Nature Communications except some minor things as followings.

(1) Some typos are located, e.g., $\text{Co}_{26}\text{F}_{54}\text{B}_{20}$ in second line of Table I, which I think should be $\text{Co}_{26}\text{Fe}_{54}\text{B}_{20}$.

Thanks for identifying this typo. We have corrected it and checked the manuscript thoroughly for additional typos.

(2) Too many papers previously published from the same group seem to be self-referenced, and I would suggest that this should be balanced. Also it is found that some papers were missing, e.g., Pushp et al, PNAS 112, 6585 (2015).

For a better balance we have added the paper by Pushp et al. and the following three papers from Jansen's group that investigate semiconductors in spin caloritronics:

1. K.-R. Jeon, H. Saito, S. Yuasa & R. Jansen Phys. Rev. B 92, 54403 (2015).
2. J. C. Le Breton, S. Sharma, H. Saito, S. Yuasa & R. Jansen Nature 475, 82–5 (2011).
3. R. Jansen Nat. Mater. 11, 400–8 (2012).

Furthermore, we have included two recent publications from T. Böhnert et al., which discuss the TMS of CoFeB based MTJs:

1. T. Böhnert et al. Phys. Rev. B 95, 104441 (2017).
2. T. Böhnert et al. J. Phys. Condens. Matter 29, 185303 (2017).

In addition, we have omitted two self-referenced publications (Liebing 2012 and Liebing 2013).

We appreciate the time and effort reviewer #3 has spent on improving our work. We hope that our corrections fully address his questions.

Reviewers' Comments:

Reviewer #1 (Remarks to the Author):

I approve the paper to be published in Nature Communications.

Reviewer #2 (Remarks to the Author):

I went through the paper and responses. I reiterate my initial recommendation to publish this paper. The authors have done a great job addressing the points of the first referee, performing new measurements supporting their conclusions. I like the paper as it is.

Reviewer #3 (Remarks to the Author):

The authors properly addressed all points raised by reviewers thus revising the paper accordingly. So I suggest that this paper be published as is.